# Validation of a risk-prediction model for pediatric post-discharge mortality after hospital admission for infection in Rwanda: A prospective cohort study

Anneka Hooft[1]☯*, Christian Umuhoza[2,3]☯, Jessica Trawin[4], Cynthia Mfuranziza[5], Emmanuel Uwiragiye[6], Cherri Zhang[4], Vuong Nguyen[4], Aaron Kornblith[1], Nathan Kenya Mugisha[7], J. Mark Ansermino[4], Matthew O. Wiens[4,7,8]☯

**1** University of California San Francisco Departments of Emergency Medicine and Pediatrics, San Francisco, California, United States of America, **2** University Teaching Hospital of Kigali, Kigali, Rwanda, **3** University of Rwanda, College of Medicine and Health Sciences, Department of Pediatrics, Kigali, Rwanda, **4** Institute for Global Health, BC Childrens' and Womens' Hospital, Vancouver, Canada, **5** Rwanda Pediatric Association, Kigali, Rwanda, **6** Ruhengeri Regional Referral Hospital, Ruhengeri, Rwanda, **7** Walimu (Uganda), Kampala, Uganda, **8** Department of Anesthesiology, Pharmacology and Therapeutics, University of British Columbia, Vancouver, British Columbia, Canada

☯ These authors contributed equally to this work.
* Anneka.hooft@ucsf.edu (AH); matthew.wiens@bcchr.ca (MW)

## Abstract

Mortality following hospital discharge remains a significant threat to child health, particularly in resource-limited settings. The Smart Discharges risk-prediction models use simple clinical, socio-behavioral, and point-of-care lab test variables to successfully predict children at the highest risk of death after hospital admission for infection to guide a risk-based approach to post-discharge care. In Rwanda, we externally validated five models derived from the prior Smart Discharges Uganda studies in a new cohort of children ages 0 days to 60 months admitted for suspected sepsis at two hospitals. We evaluated model performance using metrics including area under the receiver operating characteristic curve (AUROC), Brier score, and test characteristics (e.g., sensitivity, specificity). Performance was visualized through ROC and gain curves and calibration plots. Of 1218 total children (n = 413, Kigali; n = 805, Ruhengeri), 1161 lived to discharge (95.3%) and 1123 of those completed 6-month follow-up (96.7%). The overall rate of post-discharge mortality was 4.8% (n = 58). All five prediction models tested achieved an area under the receiver-operating curve (AUROC) greater than 0.7 (range 0.706 - 0.738). Low outcome rates resulted in moderately wide confidence intervals. Model degradation ranged from 1.1% to 7.7%, as determined by the percent reduction in AUROC between the internal validation of the original Ugandan cohort and the external Rwandan cohort. Calibration plots showed good calibration for all models at predicted probabilities below 10%. There were too few outcomes to assess calibration among those at the highest predicted risk levels.

**Data availability statement:** Study materials (protocol, consent forms, data collection tools, and metadata) are publicly available through the Pediatric Sepsis Data CoLaboratory's (Sepsis CoLab) Dataverse on Borealis, the Canadian Dataverse Repository [1]. Due to the sensitive nature of clinical data and the potential risk for re-identification of research participants, the de-identified dataset is available through moderated access in accordance with the regulation of the reviewing ethical boards [2]. Study materials including the protocol, consent forms, data collection tools, code, and metadata are publicly available via the Pediatric Sepsis Data CoLaboratory (Sepsis CoLab) Dataverse on Borealis, the Canadian Dataverse Repository. Data Access Contact Information: Pediatric Sepsis Data Governance Committee Email: sepsiscolab@bccchr.ca Repository: https://doi.org/10.5683/SP3/JV4SOA

**Funding:** This work was funded by an Early Career Award from the Thrasher Research Fund (to AH), the University of British Columbia (to MW), and the University of California, San Francisco, Department of Emergency Medicine Global Health Section (to AH). The funders had no role in study design, data collection and analysis, decision to publish, or preparation of the manuscript.

**Competing interests:** The authors have declared that no competing interests exist.

Discrimination was good with minimal degradation of the model despite low outcome rates. Future work to assess model calibration among the highest risk groups is required to ensure models are broadly generalizable to all children with suspected sepsis in Rwanda and in similar, resource-limited settings.

## Introduction

Over 3 million children under five die per year from preventable infections, with children in Sub-Saharan Africa (SSA) disproportionately affected [3–6]. While most studies focus on preventive measures and hospital-based care, post-discharge mortality after hospital admission for sepsis remains a significant, often under-recognized contributor to pediatric mortality in low-resource settings [7]. Prior studies suggest that post-discharge death usually occurs outside of the hospital setting within the first several weeks after discharge, with comorbid conditions (e.g., HIV, anemia) and socio-behavioral factors (e.g., maternal education level, travel time to hospital) linked to increased risk [8–11]. Linking these risk patterns directly to interventions to improve survival following discharge is urgently required to achieve sustainable development goals related to child health.

Key to addressing post-discharge mortality is the ability to identify the "at risk" child. Through a risk-differentiated care approach, at-risk children can receive a more intensive and personalized approach to care at time of discharge and afterwards. In austere environments with limited healthcare resources, approaches to improve care efficiency carry significant potential to help further improve child survival. The *Smart Discharges* approach is a digital health innovation linking risk prediction to personalized post-discharge care among children who are admitted with suspected infection, which has recently demonstrated potential in reducing under-5 mortality in Uganda [12,13].

The Smart Discharges models use demographic, socio-behavioral, and clinical variables present at the time of admission to predict mortality from infectious causes within the first 6 months following discharge. These models were developed and have already been internally validated within a Ugandan context [14], In comparison to other tools for the prediction of post-discharge mortality in children, these algorithms were designed to be (1) parsimonious, using 8 or fewer predictor variables, (2) simple, using only routinely collected and readily available variables and (3) flexible, through the creation of several different models to minimize the impact of missing predictors.(12) All components of these original models can be found in Table 2.

Expanding this approach outside of Uganda requires external validation of these models in additional environments. Given its close geographic proximity to Uganda, comparable rates of under-five mortality, and interest by key stakeholders, Rwanda is an ideal location to begin expansion of the Smart Discharges program. Therefore, the purpose of this study was to externally validate the Smart Discharges algorithms within the Rwandan context.

## Methods

### Study setting and population

Patients were recruited from two sites in Rwanda: University Teaching Hospital of Kigali (CHUK) and Ruhengeri Regional Referral Hospital. University Teaching Hospital of Kigali is an urban, centrally located, academic hospital in Kigali, the capital and largest city in Rwanda with a population of over 1.2 million. The pediatric ward at CHUK admits approximately 2000 children annually and has pediatric intensive care unit (ICU) capacity. Ruhengeri is a government-funded, rural, district hospital in Musanze, the largest town in Northern Rwanda with a population of around 50,000 and a catchment of around 500,000. It has an annual pediatric admission volume of 3000 cases annually and does not have ICU capacity.

### Study design and approvals

This was a prospective observational study with patients enrolled from 13 May 2022 to 23 February 2023. Children aged >0 days to 5 years admitted for suspected (e.g., clinical diagnosis of pneumonia) or proven (e.g., malaria test positive) infection as determined by the admitting clinician at the time of admission [15] were considered for enrollment. Our prior study demonstrated that the majority of these children are considered to meet clinical criteria for sepsis [16]. Inclusion criteria included: residing within the catchment area of one of the two hospitals; provision of informed consent by the child's parent or guardian; and parent or legal guardian ability to provide contact information (phone number or address) for phone or in-person follow up. Exclusion criteria included: infants admitted via the labor ward from birth and children admitted for elective procedures, trauma, or other non-infectious indications. Participants already enrolled were not considered for subsequent enrollment.

This study was approved by the institutional review boards at University of California, San Francisco (No. 21–34663, 08-Oct-2021), University of British Columbia (No. H21-02795, 28-Jan-2022), University of Rwanda (No. 411, 30-Dec-2021), and University Teaching Hospital of Kigali (No. 005, 14-Jan-2022) and by the Rwanda Paediatrics Association. This manuscript adheres to the Transparent Reporting of a multivariable prediction model for Individual Prognosis or Diagnosis (TRIPOD) statement [17].

### Study procedures

Data collection procedures were completed in accordance with data used in the original modeling derivation cohorts completed in Uganda and have been described previously in detail [15,18,19]. All participants received routine care by the medical team during admission. Briefly, following admission and written informed consent by the child's adult caregiver, a study nurse obtained clinical information including vital signs, pulse oximetry, anthropometrics (mid-upper arm circumference (MUAC), height, and weight), and Blantyre Coma Scale (BCS). Sociodemographic variables (e.g., maternal education, water source) were collected from the child's caregiver. Age-dependent demographic variables collected at enrolment were converted to age-corrected z-scores according to the World Health Organization (WHO) Child Growth Standards. Age-corrected heart rate and respiratory rate z-scores were standardized using mean and standard deviation (SD) values from Fleming et al [20]. Age-corrected z-scores for systolic blood pressures were calculated using participants' height as previously described [15,18].

A study nurse performed a finger prick blood draw on all children. Glucose testing was performed using a point-of-care (POC) glucometer. Hemoglobin was measured using a Hemocue 301 device (Brea, CA, USA). All participants also received a malaria rapid diagnostic test (RDT). Participants without a confirmed HIV serostatus also had HIV rapid testing per national algorithms, with polymerase chain reaction (PCR) confirmation as necessary based on age and exposure.

At discharge, participants were again reviewed by a study nurse who recorded current vital signs, disposition (e.g., discharge, death, transfer, eloped, discharged against medical advice), and diagnosis(es) made by the medical team. A member of the study team contacted families with active telephone lines at 2-, 4-, and 6 months after hospital discharge.

Families with no telephone access or unresponsive to phone contact received an in-person visit at these intervals by a study nurse or field officer. During these phone calls or in-person visits, study staff recorded the child's vital status and subsequent healthcare-seeking and/or readmissions since the initial visit. For all participants who died during the follow-up period, a verbal autopsy was performed, modeled after the WHO Verbal Autopsy instrument [21].

Study data were collected on the Smart Discharges App and uploaded to a secure REDCap database hosted at the BC Children's Hospital Research Institute, Vancouver, Canada. All data collection tools are available through the Smart Discharges Dataverse [1].

## Validation of smart discharges models

Briefly, the original risk prediction models were derived as follows: potential components were chosen *a priori* through a modified Delphi process and grouped into categories. We then used elastic net regression to create 3 different model types from the pool of available variables; one model using common *clinical* variables (e.g., pulse oximetry); one model using on both *clinical and social* variables (e.g., water source); and one model using *any* predictor variable (e.g., point-of-care lab test results) to allow for flexible use during implementation pending available resources and data. We then used the weighted sums of the absolute regression of each variable to rank importance and selected the top 8 eight unique variables for inclusion in the final model. Models were limited to 8 components given the anticipated challenge of using larger models during implementation in real-world settings. Separate sets of models were developed for children under 6-months of age and those 6–60 months of age, to accommodate for age-related heterogeneity in outcome prediction. See Wiens 2024 for full details on this process [14].

Sample size was calculated assuming an 8% outcome rate to allow measurement of sensitivity (estimated at 80%) of the existing risk-stratification tool, which contains 8–10 clinical, behavioral, and social variables in two sets of models stratified by age < 6 months and age 6–60 months, to within +/-3%. A sample size of 1000 participants would give a margin of error for the outcome rate of ~2% and allow for sufficient tool validation with approximately 80% power and $p = 0.05$, assuming the tool demonstrated adequate sensitivity. This would also allow for derivation of new predictive models of 5–8 variables each using a 10 event per explanatory variable estimate for a mean absolute prediction error of 0 should the original model have performed inadequately.

Existing Smart Discharges Models from the Uganda cohort (Table 2) were applied to the Rwanda cohort to obtain a risk score using only data available at time of admission (Day 0). Coefficients for individual model components were applied to each participant's unique responses for these variables and used to calculate the predicted probability of post-discharge death (e.g., percent likelihood of death) for each participant (S1 Table). Models 1a and 1b are specific to children 0–6 months of age, with Model 1a containing easily obtained clinical variables and Model 1b containing additional social factors (Time to reach hospital) (Table 2). Models 2a-c were specific to age < 6 months to 60 months, with Model 2a containing clinical, social, and laboratory (hemoglobin) variables, Model 2b containing clinical variables, and Model 2c containing clinical and social variables (Table 2). For each model, we assessed overall performance, discrimination, and calibration [22–24]. Brier Scores were calculated to assess overall performance, ranging from 0 to 1, with values closer to 0 indicating better model fit [23]. We used area under the receiver operating curve (AUROC) to assess model discrimination, visualized with receiver operating characteristic (ROC) curves, with AUROC values closer to 1 considered to have good discrimination and AUROC closer to 0.5 indicative of poor discrimination [23]. Calibration was evaluated via calibration plots of predicted versus observed rates of post-discharge death with a slope of 1 representing perfect calibration [22]. For this analysis, a sensitivity threshold was set at 80% based on our prior work and discussions with key stakeholders and determined as the ideal cut point to balance resource prioritization while minimizing risk of a severe, high-risk outcome Positive and negative predictive values (PPV, NPV) were obtained to further evaluate the models at the set sensitivity threshold with gain curves to illustrate the percentage of the cohort needed to capture a percentage of the total number of post-discharge deaths. We investigated missing values through tabulation of each variable (distribution shown in Table 1)

**Table 1. Participant characteristics and adjusted odds ratios for the risk of post-discharge mortality.**

| Variables | N = 1127 | | Missing N (%) |
|---|---|---|---|
| | N (%)/Median (IQR) | aOR* (95% CI) | |
| **Demographics** | | | |
| **Male** (ref = female) | 676 (60.0) | 0.81 (0.47-1.39) | 0 |
| **Age, months** | 13.5 (6.1-24.7) | 0.96 (0.94-0.99) | 4 (0.4) |
| **Admission clinical assessment** | | | |
| **Time since last admission** | | | 20 (1.8) |
| Never | 683 (60.6) | reference | |
| ≤ 1 month | 233 (20.7) | 0.93 (0.46-1.86) | |
| > 1 month | 211 (18.7) | 1.63 (0.78-3.41) | |
| **Weight for age z-score** | | | 11 (1.0) |
| < -3 | 103 (9.1) | 1.76 (0.76-4.12) | |
| -3 to -2 | 116 (10.3) | 0.38 (0.18-0.82) | |
| > -2 | 908 (80.6) | reference | |
| **Length for age z-score** | | | 15 (1.3) |
| < -3 | 180 (16.0) | 2.13 (1.11-4.10) | |
| -3 to -2 | 161 (14.3) | 1.24 (0.57-2.68) | |
| > -2 | 786 (69.7) | reference | |
| **Weight for length z-score** | | | 15 (1.3) |
| < -3 | 111 (9.9) | 3.76 (1.92-7.36) | |
| -3 to -2 | 108 (9.6) | 3.16 (1.55-6.46) | |
| > -2 | 908 (80.6) | reference | |
| **$SpO_2$, %** | 95 (88-97) | 0.98 (0.95-1.02) | 11 (1.0) |
| **Temperature (°C)** | | | 9 (0.8) |
| < 36.5 | 208 (18.5) | 0.75 (0.34-1.66) | |
| 36.5-37.5 | 419 (37.2) | reference | |
| > 37.5 | 500 (44.4) | 0.67 (0.37-1.22) | |
| **Abnormal BCS** | 194 (17.2) | 1.93 (1.06-3.52) | 10 (0.9) |
| **HIV positive** | 3 (0.3) | 4.04 (0.35-46.89) | 17 (1.5) |
| **Positive malaria test** | 17 (1.5) | – | 16 (1.4) |
| **Haemoglobin, g/dl** | | | 16 (1.4) |
| No anemia: ≥ 11 | 712 (63.2) | reference | |
| Anemic: < 11 | 415 (36.8) | 1.74 (1.00-3.00) | |
| **Respiratory distress** | 228 (20.2) | 1.37 (0.76-2.48) | 10 (0.9) |
| **Maternal and Social Characteristics** | | | |
| **Time to reach hospital** | | | 11 (1.0) |
| < 30 min | 453 (40.2) | reference | |
| 30 min – 1 hour | 438 (38.9) | 1.28 (0.63-2.60) | |
| 1–2 hours | 135 (12.0) | 1.76 (0.75-4.09) | |
| > 2 hours | 101 (9.0) | 2.01 (0.86-4.71) | |
| **Maternal education** | | | 9 (0.8) |
| ≤ P3 | 151 (13.4) | reference | |
| P4 to P6 | 457 (40.7) | 0.46 (0.23-0.91) | |
| S1 to S6 | 428 (38.1) | 0.24 (0.11-0.52) | |
| > S6 | 88 (7.8) | 0.13 (0.04-0.49) | |

*(Continued)*

**Table 1.** (Continued)

| | N = 1127 | | |
|---|---|---|---|
| **Boil/disinfect/filter water** | 441 (39.1) | 0.46 (0.24-0.88) | 11 (1.0) |
| **Discharge Characteristics** | | | |
| **Discharge status** | | | 0 |
| Routine discharge | 1097 (97.3) | reference | |
| Referral (higher level of care) | 21 (1.9) | 2.77 (0.85-9.01) | |
| Unplanned discharge | 9 (0.8) | 1.57 (0.18-13.50) | |
| **Length of stay (days)** | 4 (3-8) | 1.06 (1.04-1.09) | 0 |

Note: *adjusted for age, sex, and site. No outcomes were missing. Sex, discharge status, and length of stay were the only variables with no missing data.

Abbreviations: OR, odds ratio; IQR, interquartile range; BCS, Blantyre Coma scale; HIV, human immunodeficiency virus; SpO$_2$, oxygen saturation; ref, reference.

for both the whole dataset and by age group. Missing data was assumed to be random, with this assumption confirmed using Little's test and k-nearest neighbor was used for the imputation of missing values.

## Results

A total of 1706 children were screened for inclusion, of whom 1218 were enrolled (Fig 1). Of these, 57 (4.6%) died in the hospital during the index admission, and 58 (5.1%) of those who survived hospitalization died within 6 months of discharge. Among those aged under 6 months of age at enrollment, 28 (10%) died following discharge, compared to 30 (4%) children who were aged between 6 and 60 months (about 5 years) at enrollment (Fig 1). The median time to death after discharge was 29 days (Interquartile range (IQR): 16 – 90 days). The median age of children at enrollment was 13.5 months (IQR: 6.1-24.7), the majority were male (n = 676, 60%), 103 (9.1%) were considered severely underweight (Weight of age z-score < -3) and 180 (16%) were severely stunted (height for age z-score < -3). HIV prevalence was low (n = 3, 0.3%) (Table 1). Overall mortality rate was comparable to that of the Ugandan derivation cohort (5.2% vs. 6% respectively), with comparison between the two groups provided in S2 Table. Of note, prevalence of HIV and malnutrition were notably lower in the current cohort.

Study flow chart depicting the total number of potential participants who were screened for inclusion, enrolled, survived to hospital discharge, completed 6-month follow-up, and outcomes at 6-month post-discharge follow-up stratified by age group of 0 to <6 months and 6 to <60 months.

### Model performance and validation

All five models underwent validation and demonstrated good overall performance (Table 2) with minimal degradation. All p-values for all degradation percentages were not significant at the 0.05 level. Brier Scores for both Model 1 and 2 2 for the 0–6-month group were 0.09 (Fig 2A and 2B, while for Models 1, 2, and 3 for the6–60-month group the Brier Scores were 0.03 (Fig 3A, 3B, and 3C). All five prediction models tested achieved an AUROC greater than 0.7 (range 0.706 - 0.75, 95% CI 0.6-0.84), ranging from 92% to 96% of the cross-validated AUC of the models from the derivation study (Figs 2, 3). At 80% sensitivity for the top performing model in the 0–6-month cohort (Fig 2A), corresponding to a probability threshold of 0.07, the specificity, PPV, and NPV were 0.64, 0.20 and 0.96, respectively. At 80% sensitivity for the top performing model in the 6-60m cohort (Fig 3A), corresponding to a probability threshold of 0.03, specificity, PPV and NPV were 0.64, 0.05 and 0.98, respectively (Table 2).

Calibration plots demonstrated good calibration at predicted probabilities below 10% (Figs 2, 3), encompassing the majority of enrolled children. There were too few outcomes to assess calibration at predicted risks above this level. The

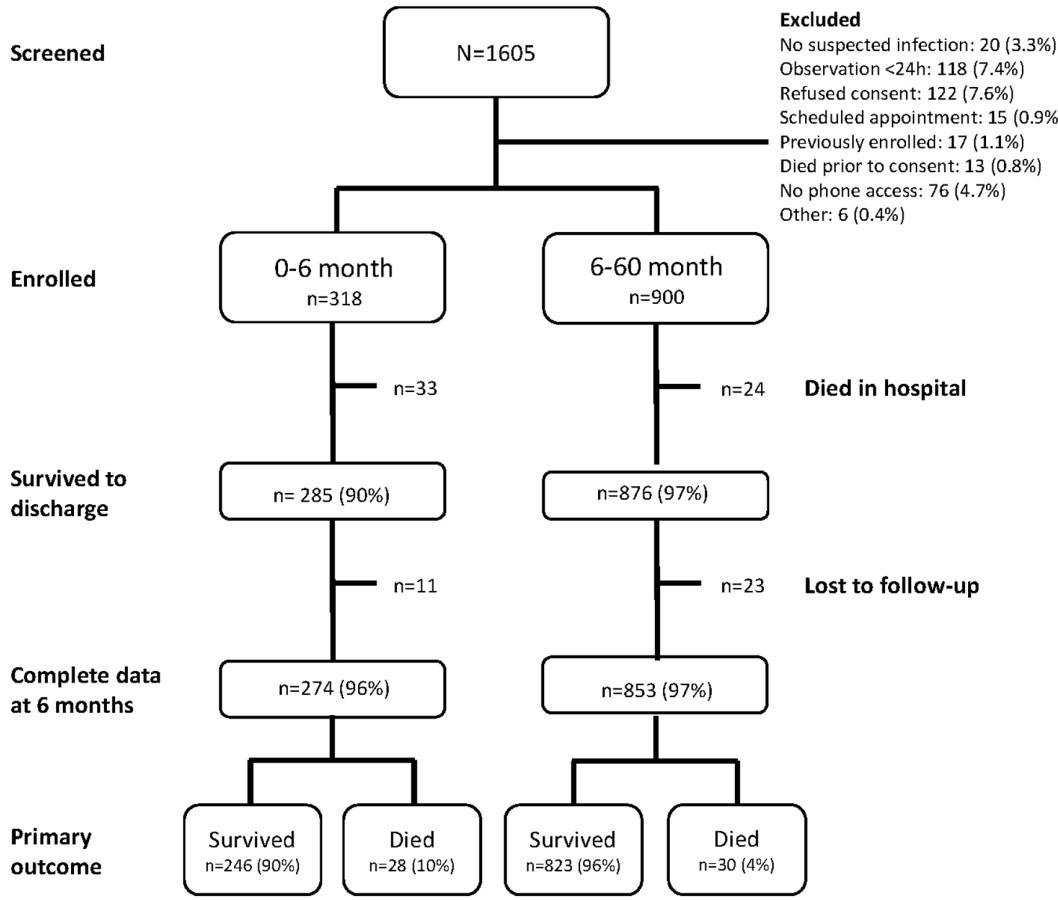

**Fig 1. Study flow chart stratified by age group.**

gain curves demonstrate that for the 0–6 month age group, selecting the top 40–45% highest risk children identified by the model in the validation cohort would identify at least 80% of all post-discharge deaths with an overall risk of mortality of 6.1% (Fig 2A) and 7.1% (Fig 2B), respectively at this threshold. In the 6–60 month group, the proportion of high-risk children needed is slightly larger at ~50% to identify 80% of all post-discharge deaths, given the lower overall risk of mortality of 3.2% (Fig 3A), 3.8% (Fig 3B), and 3.5% (Fig 3C) at this threshold, respectively.. Model performance was reduced, but remained adequate when stratified by hospital, as expected given the reduced sample size (S3–S6 Figs).

The ROC Plot (a) compares the current cohort to the original cohort in Uganda, with AUROC provided in the upper left. The Gain Curve (b) plots % mortality by % of the cohort sampled, and the point shown on the curve is the percentage of participants (starting from the highest risk) needed to be sampled (first percentage) in order to capture 80% of participants with the outcome of interest (second percentage).The calibration plot (c) provides the predicted probability (%) by Observed Event Percentage, with a Brier Score (provided in upper left) closer to 0 indicating greater model accuracy.

## Discussion

This study externally validates the Smart Discharges prediction models for post-discharge mortality in an urban and rural population in Rwanda and found that all models performed well in identifying the subset of admitted children at highest risk, where the majority of post-discharge deaths are concentrated. The ability to distinguish these high-risk children is a

**Table 2. List of variables included in each model and summary of performance at a sensitivity threshold of 0.8.**

| A) 0–6 Month Models | Model 1a | Model 1b | |
|---|---|---|---|
| **Performance characteristics** | | | |
| Specificity | 0.64 | 0.57 | |
| AUROC | 0.75 | 0.71 | |
| PPV | 0.20 | 0.17 | |
| NPV | 0.96 | 0.96 | |
| PRAUC | 0.23 | 0.25 | |
| Brier Score | 0.09 | 0.09 | |
| Model degradation (%) | 2.7 | 7.7 | |
| **Variables** | | | |
| Age, months | ✓ | ✓ | |
| Duration of present illness, categorical | ✓ | ✓ | |
| MUAC, mm | ✓ | ✓ | |
| Neonatal jaundice, binary | ✓ | ✓ | |
| Sucking well while breastfeeding, binary | ✓ | ✓ | |
| SpO2, % | ✓ | ✓ | |
| Weight for age z-score | ✓ | ✓ | |
| Fontanelle bulging, binary | ✓ | | |
| Time to reach hospital, categorical | | ✓ | |
| **B) 6–60 Month Models** | **Model 2a** | **Model 2b** | **Model 2c** |
| **Performance characteristics** | | | |
| Specificity | 0.45 | 0.55 | 0.49 |
| AUROC | 0.74 | 0.7 | 0.72 |
| PPV | 0.05 | 0.06 | 0.05 |
| NPV | 0.98 | 0.99 | 0.99 |
| PRAUC | 0.15 | 0.12 | 0.14 |
| Brier Score | 0.03 | 0.03 | 0.03 |
| Model degradation (%) | 0.8 | 4.6 | 2.8 |
| **Variables** | | | |
| Age, months | ✓ | ✓ | ✓ |
| HIV status, binary | ✓ | ✓ | ✓ |
| How long since last admission, categorical | ✓ | ✓ | ✓ |
| MUAC, mm | ✓ | ✓ | ✓ |
| SpO2, % | ✓ | ✓ | ✓ |
| Weight for age z-score | ✓ | ✓ | ✓ |
| Abnormal BCS, binary | | ✓ | |
| Respiratory rate, breaths per minute | | ✓ | |
| Temperature, °C | | ✓ | |
| Boil/disinfect/filter water | | | ✓ |
| Water source, categorical | ✓ | | ✓ |
| Hemoglobin, g/dL | ✓ | | |

Abbreviations: AUROC, area under the receiver-operating characteristic curve; PPV, positive predictive value; NPV, negative predictive value; PRAUC, precision-recall area under the curve; MUAC, mid-upper arm circumference; SpO2%, oxygen saturation; BCS, Blantyre Coma Scale.

PLOS Global Public Health

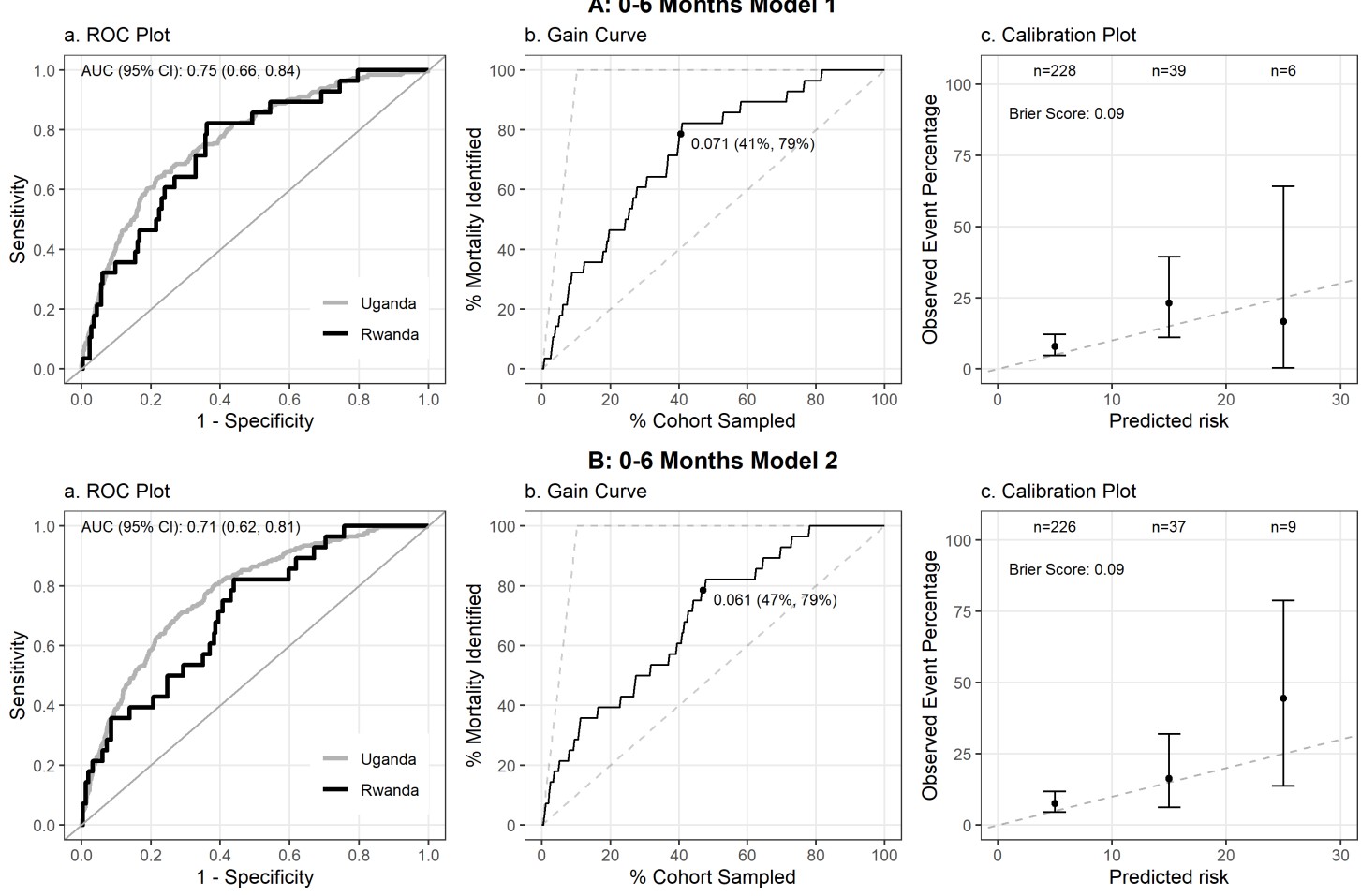

**Fig 2. Performance of the two prediction models for post-discharge mortality in Rwanda validation cohort for 0-6 month age group.**

key first step in allowing for more tailored resource allocation and prioritization. To our knowledge this is the first study to externally validate any model for pediatric post-discharge mortality within a different geographic location, health system, and timepoint. With growing global motivation to address post-discharge mortality, a risk-differentiated approach, such as that advocated by Smart Discharges in Uganda, may improve survival during the post-discharge period, particularly in Rwanda these models have now been validated.

Post-discharge mortality, though increasingly well described, has seen limited innovation in prevention, with most interventional studies not showing any benefit [25]. These have primarily included administration of prophylactic antibiotics or antimalarials and nutritional supplements for varying durations after discharge [26–31]. More comprehensive, community-based treatment programs for malnutrition have demonstrated successfully improved health outcomes, however, these have not been directly incorporated into the post-discharge follow-up process [32,33]. Among the most transformative innovations in post-discharge care interventions have been the strong findings of the mortality benefit of post-discharge malaria chemoprevention [34]. Interventions targeting health systems to improve post-discharge follow-up and engagement following discharge have not been well-explored, outside of our prior work suggesting that implementation of enhanced caregiver education upon discharge and structured follow-up in the community

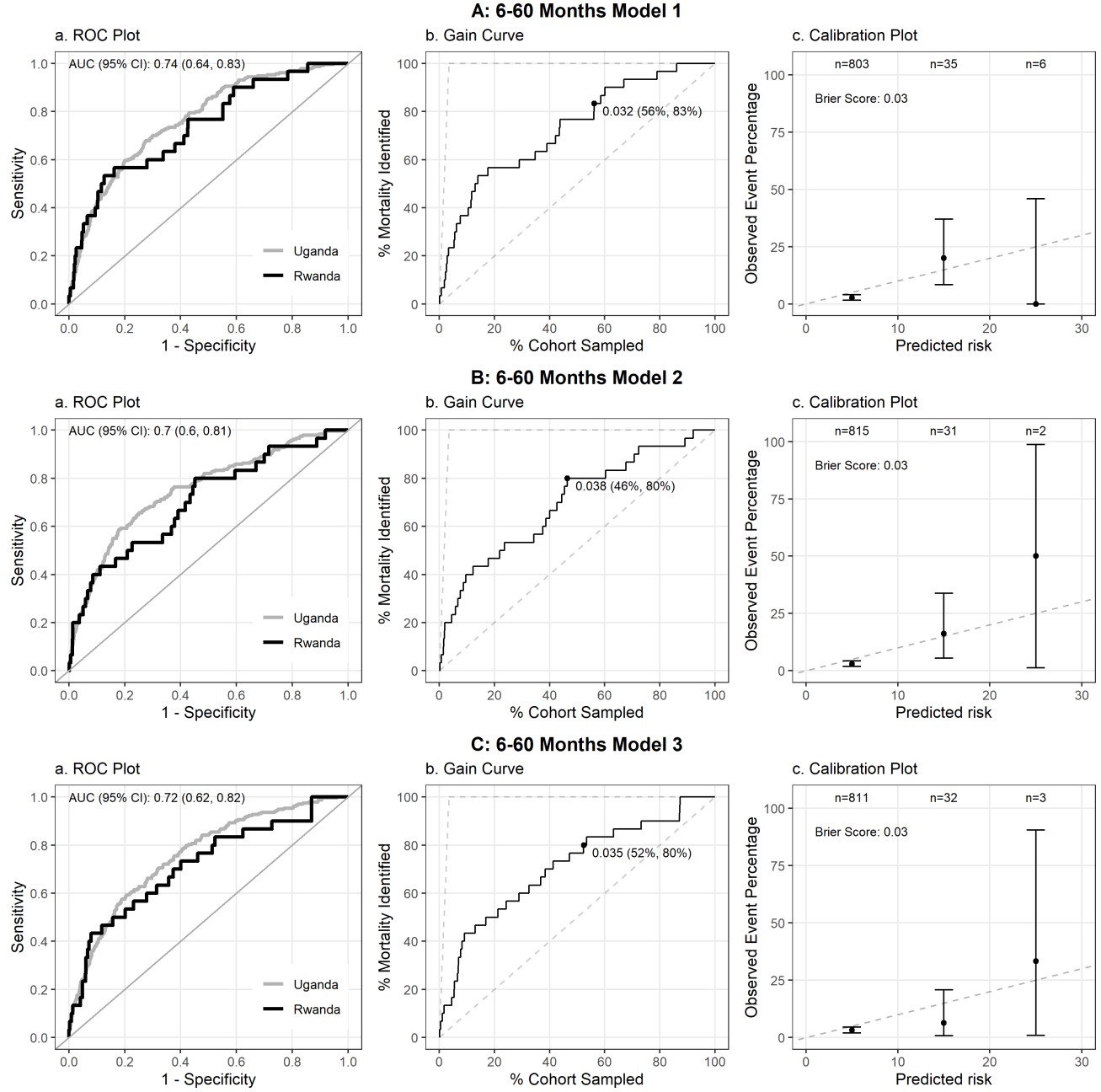

**Fig 3. Performance of the three prediction models for post-discharge mortality in Rwanda validation cohort for the 6–60-month age group.**

improve frequency of interaction with the healthcare system and likely contribute to reductions in mortality without adding significant burden on health workers or the health system [35]. The use of validated models, such as those described in this study, provide the necessary efficiency desirable for a sustainable approach to care innovations, since it would target fewer than half of the children to identify more than 80% of those likely to die. Given the serious nature of this health outcome, even in a resource-limited setting, follow-up of 50% of high-risk children is still likely to be highly feasible and cost-effective, given the possible prevention of more costly readmissions and utilization of

simple interventions such as improved caregiver education at discharge or follow-up visits at community-based health centers.

Risk prediction models are often developed but rarely validated [36–38]. Despite the proliferation of models for predicting pediatric mortality, particularly in the context of sepsis and febrile illness, the majority of those that have been externally validated were designed for use in high-resource settings using data from patients admitted to critical care units [39–47]. While some may hold promise for adaptation to lower-resource settings, risk prediction models are often not linked to care pathways that leverage the potential utility of risk prediction. This highlights the critical importance of studying these models alongside the appropriate intervention(s) to determine their impact on clinical outcomes.

External validation as described in this analysis represents only a single aspect of ensuring the continued fidelity of models within their implementation context. Alone, these findings need to provide more evidence for their broad application on an ongoing basis in Rwanda. Rather, the implementation of models using a product life-cycle approach is needed to ensure that models remain accurate over time. Health systems that incorporate models into care must ensure that all models undergo periodic re-validation and updating to avoid potential harm caused by incorrectly specified models [48]. This is analogous to post-market surveillance of pharmaceutical products or medical devices. Indeed, regulatory frameworks will soon also require researchers and innovators who develop algorithms used as decision support tools to create robust plans for monitoring and updating them before they can be translated to clinical practice [49,50]. Currently, the Smart Discharges models are ready for more large-scale implementation to evaluate their effectiveness in combination with risk-stratified prevention programs in other facilities across Rwanda. Future studies are needed to refine these models, combine them with risk-stratified prevention strategies and integrate them into clinical decision-making processes to determine their impact on post-discharge mortality in diverse settings. Furthermore, implementation outcomes such as feasibility, accessibility, fidelity, and impacts on cost and other markers of care quality and resource utilization will need to be further elucidated to understand use and optimize uptake in real world settings.

This study is subject to several limitations. First, this study observed few outcomes, limiting the inferential strengths of the validation results, thus, this may make it more difficult to rule out potentially important differences. However, this was anticipated in our sample size calculation and the results observed among two very heterogeneous facilities still suggest robustness in the findings. Second, this study had limited power to evaluate calibration at levels of risk beyond 10%, however, practically speaking, above a certain risk threshold, associated care pathways for prevention likely would not vary significantly. Calibration at low levels of risk was excellent, limiting the risk of false negative classifications, which is the primary patient-level risk in implementing risk-stratified care. Finally, these models were validated using prospective research data, which may not reflect the data captured during routine care by facility healthcare staff. In such settings, these models may reflect an optimistic scenario, both regarding data quality and completeness, hence the need for future studies assessing implementation outcomes in real-world settings. Validation using routinely captured health data would further lend confidence to these models and should be considered during any implementation scenario. In addition, cost of routine integration of risk-stratified care into clinical practice and the implications of risk reduction interventions also needs to be explored. We also acknowledge that the tool requires use of an electronic device, however, we feel this is becoming increasingly feasible as this tool can easily be utilized on a phone-based app by a provider at the bedside without need for internet connection or integrated into an electronic medical records system as more countries, including Rwanda, push toward digitalization of care.

## Conclusion

Pediatric post-discharge mortality remains a key gap in pediatric care low-resource settings. The Smart Discharges post-discharge predictive models performed well on external validation of an urban and rural population of children from Rwanda. With the validation of five simple prediction models, efforts to build risk-differentiated care pathways in Rwanda are now possible for children with suspected sepsis.

## Supporting information

**S1 Table. List of all Smart Discharges Model variates and coefficients.**
(DOCX)

**S2 Table. Comparison of baseline demographics for Rwanda validation cohort and Uganda original derivation cohort.**
(DOCX)

**S3 Table. Rwandan validation cohort characteristics, participant disposition, and post-discharge outcomes at 6 months, stratified by site.**
(DOCX)

**S1 Checklist. TRIPOD Checklist.**
(DOCX)

**S1 Fig. Hazard Curve for post-discharge mortality for the Rwanda validation cohort.**
(TIFF)

**S2 Fig. Full validation profile for 0–6 m Model 1.** The ROC Plot (a) with the probability of mortality given for the peak AUR; the precision-recall (PR) Plot (b) of model sensitivity by positive predictive value (PPV) to illustrate identification of relevant cases; Probability Thresholds (c) plots outcome probability by sensitivity and specificity, respectively. The Gain Curve (d) depicts % mortality by % of the cohort sampled, and the point shown on the curve is the percentage of participants (starting from the highest risk) needed to be sampled (first percentage) in order to capture 80% of participants with the outcome of interest (second percentage).The calibration plot (e) provides the predicted probability (%) by Observed Event Percentage, with a Brier Score (provided in upper left) closer to 0 indicating greater model accuracy; Predicted Probabilities (f) demonstrates the distribution of predicted probabilities for death and survival given by the model.
(TIFF)

**S3 Fig. Full validation profile for 0-6m Model 2.** The ROC Plot (a) with the probability of mortality given for the peak AUR; the precision-recall (PR) Plot (b) of model sensitivity by positive predictive value (PPV) to illustrate identification of relevant cases; Probability Thresholds (c) plots outcome probability by sensitivity and specificity, respectively. The Gain Curve (d) depicts % mortality by % of the cohort sampled, and the point shown on the curve is the percentage of participants (starting from the highest risk) needed to be sampled (first percentage) in order to capture 80% of participants with the outcome of interest (second percentage).The calibration plot (e) provides the predicted probability (%) by Observed Event Percentage, with a Brier Score (provided in upper left) closer to 0 indicating greater model accuracy; Predicted Probabilities (f) demonstrates the distribution of predicted probabilities for death and survival given by the model.
(TIFF)

**S4 Fig. Full validation profile for 6-60m Model 1.** The ROC Plot (a) with the probability of mortality given for the peak AUR; the precision-recall (PR) Plot (b) of model sensitivity by positive predictive value (PPV) to illustrate identification of relevant cases; Probability Thresholds (c) plots outcome probability by sensitivity and specificity, respectively. The Gain Curve (d) depicts % mortality by % of the cohort sampled, and the point shown on the curve is the percentage of participants (starting from the highest risk) needed to be sampled (first percentage) in order to capture 80% of participants with the outcome of interest (second percentage).The calibration plot (e) provides the predicted probability (%) by Observed Event Percentage, with a Brier Score (provided in upper left) closer to 0 indicating greater model accuracy; Predicted Probabilities (f) demonstrates the distribution of predicted probabilities for death and survival given by the model.
(TIFF)

**S5 Fig. Full validation profile for 6-60m Model 2.** The ROC Plot (a) with the probability of mortality given for the peak AUR; the precision-recall (PR) Plot (b) of model sensitivity by positive predictive value (PPV) to illustrate identification of relevant cases; Probability Thresholds (c) plots outcome probability by sensitivity and specificity, respectively. The Gain Curve (d) depicts % mortality by % of the cohort sampled, and the point shown on the curve is the percentage of participants (starting from the highest risk) needed to be sampled (first percentage) in order to capture 80% of participants with the outcome of interest (second percentage).The calibration plot (e) provides the predicted probability (%) by Observed Event Percentage, with a Brier Score (provided in upper left) closer to 0 indicating greater model accuracy; Predicted Probabilities (f) demonstrates the distribution of predicted probabilities for death and survival given by the model. (TIFF)

**S6 Fig. Full validation profile for 6-60m Model 3.** The ROC Plot (a) with the probability of mortality given for the peak AUR; the precision-recall (PR) Plot (b) of model sensitivity by positive predictive value (PPV) to illustrate identification of relevant cases; Probability Thresholds (c) plots outcome probability by sensitivity and specificity, respectively. The Gain Curve (d) depicts % mortality by % of the cohort sampled, and the point shown on the curve is the percentage of participants (starting from the highest risk) needed to be sampled (first percentage) in order to capture 80% of participants with the outcome of interest (second percentage).The calibration plot (e) provides the predicted probability (%) by Observed Event Percentage, with a Brier Score (provided in upper left) closer to 0 indicating greater model accuracy; Predicted Probabilities (f) demonstrates the distribution of predicted probabilities for death and survival given by the model. (TIFF)

**S7 Fig. Validation profile for 0-6m Model 1 in Kigali subset.** The ROC Plot (a) with the probability of mortality given for the peak AUR; the precision-recall (PR) Plot (b) of model sensitivity by positive predictive value (PPV) to illustrate identification of relevant cases; Probability Thresholds (c) plots outcome probability by sensitivity and specificity, respectively. The Gain Curve (d) depicts % mortality by % of the cohort sampled, and the point shown on the curve is the percentage of participants (starting from the highest risk) needed to be sampled (first percentage) in order to capture 80% of participants with the outcome of interest (second percentage).The calibration plot (e) provides the predicted probability (%) by Observed Event Percentage, with a Brier Score (provided in upper left) closer to 0 indicating greater model accuracy; Predicted Probabilities (f) demonstrates the distribution of predicted probabilities for death and survival given by the model. (TIFF)

**S8 Fig. Validation profile for 0-6m Model 1 in Ruhengeri subset.** The ROC Plot (a) with the probability of mortality given for the peak AUR; the precision-recall (PR) Plot (b) of model sensitivity by positive predictive value (PPV) to illustrate identification of relevant cases; Probability Thresholds (c) plots outcome probability by sensitivity and specificity, respectively. The Gain Curve (d) depicts % mortality by % of the cohort sampled, and the point shown on the curve is the percentage of participants (starting from the highest risk) needed to be sampled (first percentage) in order to capture 80% of participants with the outcome of interest (second percentage).The calibration plot (e) provides the predicted probability (%) by Observed Event Percentage, with a Brier Score (provided in upper left) closer to 0 indicating greater model accuracy; Predicted Probabilities (f) demonstrates the distribution of predicted probabilities for death and survival given by the model. (TIFF)

**S9 Fig. Validation profile for 6-60m Model 1 in Kigali subset.** The ROC Plot (a) with the probability of mortality given for the peak AUR; the precision-recall (PR) Plot (b) of model sensitivity by positive predictive value (PPV) to illustrate identification of relevant cases; Probability Thresholds (c) plots outcome probability by sensitivity and specificity, respectively. The Gain Curve (d) depicts % mortality by % of the cohort sampled, and the point shown on the curve is the percentage of participants (starting from the highest risk) needed to be sampled (first percentage) in order to capture 80% of participants

with the outcome of interest (second percentage).The calibration plot (e) provides the predicted probability (%) by Observed Event Percentage, with a Brier Score (provided in upper left) closer to 0 indicating greater model accuracy; Predicted Probabilities (f) demonstrates the distribution of predicted probabilities for death and survival given by the model. (TIFF)

**S10 Fig. Validation profile for 6-60m Model 1 in Ruhengeri subset.** The ROC Plot (a) with the probability of mortality given for the peak AUR; the precision-recall (PR) Plot (b) of model sensitivity by positive predictive value (PPV) to illustrate identification of relevant cases; Probability Thresholds (c) plots outcome probability by sensitivity and specificity, respectively. The Gain Curve (d) depicts % mortality by % of the cohort sampled, and the point shown on the curve is the percentage of participants (starting from the highest risk) needed to be sampled (first percentage) in order to capture 80% of participants with the outcome of interest (second percentage).The calibration plot (e) provides the predicted probability (%) by Observed Event Percentage, with a Brier Score (provided in upper left) closer to 0 indicating greater model accuracy; Predicted Probabilities (f) demonstrates the distribution of predicted probabilities for death and survival given by the model. Abbreviations: ROC: receiver-operator characteristic, AUC: area under the curve, PPV: positive predictive value, NPV: negative predictive value, PR: Precision-recall.
(TIFF)

## Acknowledgments

We would like to acknowledge all past and present members of the Smart Discharges Research program for their efforts in data collection, administration, logistics support, and all study activities, including but not limited to: Godfroid Rucinga, Esperence Umulisa, Didas Mugambinumwe, Jeanne d'Ark Mazimpaka, Claudine Uwingabiye, Theogene Bizimungu, Juliette Unyuzumutima, Peter Lewis, and Martina Knappett.

## Author contributions

**Conceptualization:** Anneka Hooft, Christian Umuhoza, Emmanuel Uwiragiye, Nathan Kenya Mugisha, J Mark Ansermino, Matthew O. Wiens.

**Data curation:** Anneka Hooft, Christian Umuhoza, Jessica Trawin, Cherri Zhang, Vuong Nguyen, Matthew O. Wiens.

**Formal analysis:** Anneka Hooft, Cherri Zhang, Vuong Nguyen, Matthew O. Wiens.

**Funding acquisition:** Anneka Hooft, Jessica Trawin, J Mark Ansermino, Matthew O. Wiens.

**Investigation:** Anneka Hooft, Christian Umuhoza, Cynthia Mfuranziza, Emmanuel Uwiragiye, Aaron Kornblith, Matthew O. Wiens.

**Methodology:** Anneka Hooft, Christian Umuhoza, Jessica Trawin, Emmanuel Uwiragiye, Cherri Zhang, Vuong Nguyen, Aaron Kornblith, Nathan Kenya Mugisha, Matthew O. Wiens.

**Project administration:** Anneka Hooft, Christian Umuhoza, Jessica Trawin, Cynthia Mfuranziza, Emmanuel Uwiragiye, J Mark Ansermino, Matthew O. Wiens.

**Resources:** Cynthia Mfuranziza, Nathan Kenya Mugisha, Matthew O. Wiens.

**Software:** Matthew O. Wiens.

**Supervision:** Anneka Hooft, Christian Umuhoza, Jessica Trawin, Emmanuel Uwiragiye, Vuong Nguyen, Aaron Kornblith, J Mark Ansermino, Matthew O. Wiens.

**Validation:** Anneka Hooft, Christian Umuhoza, Cherri Zhang, Vuong Nguyen, Aaron Kornblith, Matthew O. Wiens.

**Visualization:** Anneka Hooft, Cherri Zhang, Vuong Nguyen, Matthew O. Wiens.

**Writing – original draft:** Anneka Hooft, Christian Umuhoza, Matthew O. Wiens.

**Writing – review & editing:** Anneka Hooft, Christian Umuhoza, Jessica Trawin, Cynthia Mfuranziza, Emmanuel Uwiragiye, Cherri Zhang, Vuong Nguyen, Aaron Kornblith, Nathan Kenya Mugisha, J Mark Ansermino, Matthew O. Wiens.

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
