## [Decision Letter · Decision Letter 0]

PGPH-D-24-02666

Validation of a risk-prediction model for pediatric post-discharge mortality after hospital admission in Rwanda

Dear Dr. Hooft,

Thank you for submitting your manuscript to PLOS Global Public Health. After careful consideration, we feel that it has merit but does not fully meet PLOS Global Public Health’s publication criteria as it currently stands. Therefore, we invite you to submit a revised version of the manuscript that addresses the points raised during the review process.

The manuscript has been evaluated by two reviewers, and their comments are available below.

Could you please carefully revise the manuscript to address all comments raised?

We look forward to receiving your revised manuscript.

Kind regards,

Jianhong Zhou

Staff Editor

Journal Requirements:

Additional Editor Comments (if provided):

Reviewers' comments:

Reviewer's Responses to Questions

**Comments to the Author**

1. Does this manuscript meet PLOS Global Public Health’s publication criteria?

Reviewer #1: Yes

Reviewer #2: Partly

2. Has the statistical analysis been performed appropriately and rigorously?

Reviewer #1: Yes

Reviewer #2: Yes

3. Have the authors made all data underlying the findings in their manuscript fully available (please refer to the Data Availability Statement at the start of the manuscript PDF file)?

Reviewer #1: No

Reviewer #2: Yes

4. Is the manuscript presented in an intelligible fashion and written in standard English?

Reviewer #1: Yes

Reviewer #2: Yes

Reviewer #1: Thank you for the opportunity to review this manuscript. The authors present a well-designed and clearly written validation study that makes an important contribution to the field of post-discharge management in low-resource settings. However, the introduction and the methods section is written too specific to the Smart discharge study and suggests that the reader is expected to know about this other project in advance. Please elaborate throughout the manuscript on the Smart discharge programme and the models used. The major comments are indicated below and ought to be addressed before publication.

Major comments

Abstract:

line 27. It would more informative for the reader if the Smart discharge risk prediction model is explained more generally. Currently it assumes the reader knows what the author's mean by Smart discharge.

line 29. The methods in this section need to be explained more. Currently all we know from the abstract is that the models were validated, but the statistical methods are unknown.

Introduction:

line 54. It would be useful to briefly describe in more depth the "socio-behavioural factors linked to increased risk".

line 65. Please elaborate what we mean by variables present on admission? are these signs and symptoms or socio-demographic variables?

Methods

Line 90. The criteria for admission needs greater clarity. Please elaborate on "suspected or proven infection".

Validation of Smart Discharges Models

line 144. There no mention of the types of prediction models used in this study. A brief description is essential to understand the performance of the model.

similarly, it is not clear which variables were used in the models from the text. One has to review the supplementary material to ascertain this.

line 159. Please describe the justification for the 80% cut-off in more depth.

line 162. Please describe more fully how missing data was investigated. What was the distribution of missing variables more generally and were these assumed to be missing at random?

Similarly, was this done before or after the data were split.

Results

Table 1 and Table 2 appear to be missing from the document.

line 179. Please consider including confidence intervals for the model degradation percentages to better contextualize the findings

The sample size was estimated at 8% post-discharge mortality rate, but the observed rate was lower (4.8%), which may have impacted the study’s power. Please elaborate on this in the discussion.

Discussion:

The discussion would benefit from more detailed exploration of how these models could be integrated into routine clinical care, particularly given the authors' appropriate concerns about data quality when collected outside research settings.

Some consideration of the cost implications of implementing risk-stratified care would be valuable.

Figures and Tables

Fig 1 and 2, could benefit from clearer labelling and captions to enhance accessibility for non-specialist readers. Ideally these should stand alone.

Reviewer #2: Unfortunately, I did not have access to Tables 1 and 2 in the documents submitted (only the figures and supplementary material were available), so some of my comments may not be relevant.

--

This article describes the external validation of a model predicting post-discharge mortality in children admitted with suspected sepsis. The model was previously developed on a cohort of children in Uganda, and is validated here on a cohort of children admitted to 2 Rwandan hospitals. The article is interesting and well-written.

However, there are several important elements currently missing from the manuscript, including:

1.In the title: I recommend to mention that the model only concerns post-discharge mortality for suspected sepsis (not all cause hospitalisations).

2. Line 90: ‘Suspected or proven infection’: it would be useful to specify which criteria define a ‘suspected or proven infection’, who assesses them (clinician or researcher), and when (on admission, during hospitalisation). Furthermore, line 63 mentions that the model was developed on children admitted with suspected sepsis, and not all children with ‘suspected or proven infection’. I would clarify/rationale the inclusion criteria.

3. Clarify whether only day 0 data were included in the score (since vitals/other parameters at discharge were captured)

4. Line 144: Is there a reference detailing the initial model development? In particular, it would be useful to know how the various predictors are integrated into the initial model (continuous, categorical (if so, number and definitions of categories)).

5.Line 144 : Reference to Table 1 seems to be erroneous (title does not correspond to line 176)

6.Line 145 : Explain how the risk score is used to calculate the predicted probability of death

7.Results : The proportion of missing data for predictors and outcome are not mentioned. Since the k-nearest neighbor method was used for the imputation of missing values, were the missing data missing at random?

8. Line 184 : Why the specificity is not mentioned ?

9. The study was carried out in 2 different HFs, a university hospital and a district hospital: mortality is probably higher in the university hospital. Would it therefore be possible to specify the results of the primary outcome depending on the hospital? [Comment to be ignored if present in Table 2]. It may also be interesting to have a sensitivity analysis of the model's performance on these sub-populations (Kigali University Hospital versus regional hospital), as the performance may be different.

10. Comparison with the development data of the distribution of important variables (demographics, some important factors such as HIV prevalence, predictors and outcome) is missing. It would also have been interesting to mention the type of population on which the Ugandan model was developed (was it a population from district/regional/zonal hospitals), and to compare it with the validation population in Rwanda.

11.Discussion: It seems important to discuss the low positive predictive value at 6-60 months, which implies many ‘false positives’: discuss the potential impact in term of resources to have to provide additional follow-up for 50% of discharged children in a context of limited resources.

12. Line 252: discuss the low number of patients with outcomes (n=58) (below the common rule of thumb to have at least 100 patients with the outcome of interest in the validation dataset to ensure reliable estimates of model performance + below the number estimated in the sample size calculation)

13. Limitation: Consider mentioning that it is not a bedside model (not easy-to-use simplified model-requires an electronic device)

14. Fig 2: Rephrase ‘bin midpoint’ on the x-axis

**Do you want your identity to be public for this peer review?** For information about this choice, including consent withdrawal, please see our Privacy Policy

Reviewer #1: **Yes: ** Dr Sham Lal

Reviewer #2: No

---

## [Decision Letter · Decision Letter 1]

PGPH-D-24-02666R1

Validation of a risk-prediction model for pediatric post-discharge mortality after hospital admission for infection in Rwanda

Dear Dr. Hooft,

Thank you for submitting your manuscript to PLOS Global Public Health. After careful consideration, we feel that it has merit but does not fully meet PLOS Global Public Health’s publication criteria as it currently stands. Therefore, we invite you to submit a revised version of the manuscript that addresses the points raised during the review process.

Following review of your revised submission, both reviewers felt that substantial improvements have been made and requested additional changes to improve your manuscript before it can be considered for publication. Please carefully address all reviewer requests and clearly respond to each in your resubmission.

We look forward to receiving your revised manuscript.

Kind regards,

Jennifer Tucker, PhD

Staff Editor

Journal Requirements:

1. Please include a complete copy of PLOS’ questionnaire on inclusivity in global research in your revised manuscript. Our policy for research in this area aims to improve transparency in the reporting of research performed outside of researchers’ own country or community. The policy applies to researchers who have travelled to a different country to conduct research, research with Indigenous populations or their lands, and research on cultural artefacts. The questionnaire can also be requested at the journal’s discretion for any other submissions, even if these conditions are not met.  Please find more information on the policy and a link to download a blank copy of the questionnaire here: https://journals.plos.org/globalpublichealth/s/best-practices-in-research-reporting. Please upload a completed version of your questionnaire as Supporting Information when you resubmit your manuscript.”

2. Please note that PLOS Global Public Health has specific guidelines on code sharing for submissions in which author-generated code underpins the findings in the manuscript. In these cases, all author-generated code must be made available without restrictions upon publication of the work. Please review our guidelines at https://journals.plos.org/globalpublichealth/s/materials-and-software-sharing#loc-sharing-code and ensure that your code is shared in a way that follows best practice and facilitates reproducibility and reuse.

Additional Editor Comments (if provided):

Reviewers' comments:

Reviewer's Responses to Questions

**Comments to the Author**

Reviewer #1: (No Response)

Reviewer #2: (No Response)

publication criteria?

Reviewer #1: Yes

Reviewer #2: Yes

3. Has the statistical analysis been performed appropriately and rigorously?

Reviewer #1: Yes

Reviewer #2: Yes

4. Have the authors made all data underlying the findings in their manuscript fully available (please refer to the Data Availability Statement at the start of the manuscript PDF file)?

Reviewer #1: No

Reviewer #2: Yes

5. Is the manuscript presented in an intelligible fashion and written in standard English?

Reviewer #1: Yes

Reviewer #2: Yes

Reviewer #1: Thank you for the opportunity to review the manuscript again. The authors have made a number of substantial improvements in clarity and reporting of the study. However there are still areas where the reporting could be strengthened further. I have listed these as major comments below.

Major comments

Title: Please state the study design in the title. I understand this to be a prospective cohort study.

Introduction lines 80-94: It appears the authors have included a detailed set methods in this section. Please move this to the methods section and reserve the introduction for describing the current state of knowledge related to this research question and the added value this study generates.

Study design:

line 112: "Children aged 0 days... ". It seems strange to include children not yet born for enrolment. I think the authors mean >0 days".

line 127: Please add the TRIPOD checklist and reference in the text. This can be found in table 1 in the cited reference.

Tables and figures

These need to stand alone from the main text for interpretation. For example, the figure caption for table 1 is very vague. Please elaborate the description of the table more fully for the reader. Check and revise all tables and figures.

Table 1: State the units of time clearly. For example, Time since last admission (m) could be minutes or months?

Table 1: Length of stay what are the units?

Minor comments

Abstract: The abstract names Uganda in the first section multiple times and then Rwanda. It might be confusing to reader, as to which country this study was conducted in and relevant for.

Reviewer #2: Thank you for the revised manuscript and the detailed responses to the comments. The authors have appropriately addressed the points raised and provided clear and satisfactory answers.

I have one additional suggestion regarding my earlier comment:

“The study was conducted in two different hospitals, a university hospital and a regional hospital: mortality is probably higher in the university hospital. Would it therefore be possible to specify the results of the primary endpoint according to the hospital? [Comment to be ignored if included in Table 2]. It might also be interesting to perform a sensitivity analysis of the model's performance on these subpopulations (Kigali university hospital versus regional hospital), as performance may differ.”

Thank you for providing the model performance stratified by facility. As a complement, I recommend adding (in the supplementary material) a table presenting the baseline characteristics stratified by hospital type (university vs. district-level). In addition, please indicate whether the outcome differed between the two settings.

This information would help clarify potential context-specific differences and, together with the stratified model performance metrics, support the assessment of the model’s generalizability.

**Do you want your identity to be public for this peer review?** For information about this choice, including consent withdrawal, please see our Privacy Policy

Reviewer #1: **Yes: ** Dr Sham Lal PhD

Reviewer #2: No

---

## [Decision Letter · Decision Letter 2]

Validation of a risk-prediction model for pediatric post-discharge mortality after hospital admission for infection in Rwanda: a prospective cohort study

PGPH-D-24-02666R2

Dear Dr. Hooft,

We are pleased to inform you that your manuscript 'Validation of a risk-prediction model for pediatric post-discharge mortality after hospital admission for infection in Rwanda: a prospective cohort study' has been provisionally accepted for publication in PLOS Global Public Health.

Best regards,

Julia Robinson

Executive Editor

Reviewer Comments (if any, and for reference):

Reviewer's Responses to Questions

**Comments to the Author**

Reviewer #1: All comments have been addressed

Reviewer #2: All comments have been addressed

publication criteria?

Reviewer #1: Yes

Reviewer #2: (No Response)

3. Has the statistical analysis been performed appropriately and rigorously?

Reviewer #1: Yes

Reviewer #2: (No Response)

4. Have the authors made all data underlying the findings in their manuscript fully available (please refer to the Data Availability Statement at the start of the manuscript PDF file)?

Reviewer #1: No

Reviewer #2: (No Response)

5. Is the manuscript presented in an intelligible fashion and written in standard English?

Reviewer #1: Yes

Reviewer #2: (No Response)

Reviewer #1: (No Response)

Reviewer #2: (No Response)

**Do you want your identity to be public for this peer review?** For information about this choice, including consent withdrawal, please see our Privacy Policy

Reviewer #1: **Yes: ** Dr Sham Lal

Reviewer #2: No
